# Cytological Diagnosis of Classic Myeloproliferative Neoplasms at the Age of Molecular Biology

**DOI:** 10.3390/cells12060946

**Published:** 2023-03-20

**Authors:** Sophie Combaluzier, Julie Quessada, Norman Abbou, Robin Arcani, Antoine Tichadou, Jean Gabert, Régis Costello, Marie Loosveld, Geoffroy Venton, Yaël Berda-Haddad

**Affiliations:** 1Hematology Laboratory, Timone University Hospital, 13005 Marseille, France; 2Hematological Cytogenetics Laboratory, Timone University Hospital, 13005 Marseille, France; 3CNRS, INSERM, CIML, Luminy Campus, Aix-Marseille University, 13009 Marseille, France; 4Molecular Biology Laboratory, North University Hospital, 13015 Marseille, France; 5INSERM, INRAE, C2VN, Luminy Campus, Aix-Marseille University, 13005 Marseille, France; 6Department of Internal Medicine, Timone University Hospital, 13005 Marseille, France; 7Hematology and Cellular Therapy Department, Conception University Hospital, 13005 Marseille, France; 8TAGC, INSERM, UMR1090, Luminy Campus, Aix-Marseille University, 13005 Marseille, France

**Keywords:** myeloproliferative neoplasms, cytomorphology, molecular biology, laboratory practice

## Abstract

Myeloproliferative neoplasms (MPN) are clonal hematopoietic stem cell-derived disorders characterized by uncontrolled proliferation of differentiated myeloid cells. Two main groups of MPN, *BCR::ABL1*-positive (Chronic Myeloid Leukemia) and *BCR::ABL1*-negative (Polycythemia Vera, Essential Thrombocytosis, Primary Myelofibrosis) are distinguished. For many years, cytomorphologic and histologic features were the only proof of MPN and attempted to distinguish the different entities of the subgroup *BCR::ABL1*-negative MPN. World Health Organization (WHO) classification of myeloid neoplasms evolves over the years and increasingly considers molecular abnormalities to prove the clonal hematopoiesis. In addition to morphological clues, the detection of JAK2, MPL and CALR mutations are considered driver events belonging to the major diagnostic criteria of *BCR::ABL1*-negative MPN. This highlights the preponderant place of molecular features in the MPN diagnosis. Moreover, the advent of next-generation sequencing (NGS) allowed the identification of additional somatic mutations involved in clonal hematopoiesis and playing a role in the prognosis of MPN. Nowadays, careful cytomorphology and molecular biology are inseparable and complementary to provide a specific diagnosis and to permit the best follow-up of these diseases.

## 1. Introduction

Myeloproliferative neoplasms (MPN) result in excessive cellular proliferation related to clonal alterations of hematopoietic stem cells (HSC). For many years, cytomorphologic and histologic features were the only proof of MPN. The World Health Organization (WHO) classification of myeloid neoplasms evolves over the years with an integrated approach for precise classification. Molecular abnormalities take the preponderant place in the diagnosis to prove the clonal origin of the myeloproliferation, to improve risk stratification and to assess prognosis.

A morphological description of a large category of ‘Myeloproliferative Disorders’ was made for the first time in 1951 by the hematologist William Dameshek who described Chronic Granulocytic Leukemia, Polycythemia Vera (PV), Megakaryocytic Leukemia, Idiopathic Myeloid Metaplasia and Erythroleukemia [1]. In 1960, an abnormally small Y-like chromosome was accidentally discovered by Nowell and Hungerford in Pennsylvania in patients with Chronic Granulocytic Leukemia and was named “Philadelphia chromosome” (Ph). It was only identified in 1973 as a reciprocal translocation between chromosomes 9 and 22 which results in the fusion of two genes: Abelson (*ABL1*) and Breakpoint Cluster Region (*BCR*), respectively, located in chromosome 9 and 22 [2,3]. This fusion gene *BCR::ABL1* was studied for years in Chronic Myeloid Leukemia (CML) and permitted to explain its physiopathology. *BCR::ABL1*-positive cells are characterized by an excessive proliferative capacity conferred by the constitutive activation of the BCR-ABL tyrosine kinase protein which in turn activates several signaling pathways including JAK/STAT, PI3K/AKT and RAS/MAPK involved in cell growth, cell survival, cellular transformation and inhibition of apoptosis [4,5,6]. It was a model in cancer disease and contributed to developing specific therapy with Tyrosine Kinase Inhibitors (TKIs) which has modified the course of the disease until now. For a long time, the other myeloproliferative disorders were defined as *BCR::ABL1*-negative MPN in successive classifications.

In the 2008 WHO classification, myeloproliferative disorders initially called ‘Chronic MyeloProliferative Diseases’ (CMPD) were reclassified as ‘MyeloProliferative Neoplasms’ (MPN) with nine entities and were split into *BCR::ABL1*-positive MPN (CML) and *BCR::ABL1*-negative MPN. Moreover, among these, the distinction was made between ‘classical *BCR::ABL1*-negative MPN’, characterized by a Janus Kinase (JAK) protein deregulation and including Polycythemia Vera (PV), Essential Thrombocythemia (ET), Primary Myelofibrosis (PMF) and other *BCR::ABL1*-negative MPN including Chronic Neutrophilic Leukemia (CNL), Chronic Eosinophilic Leukemia (CEL), not otherwise specified, Hypereosinophilic Syndrome (HES), Chronic myeloproliferative neoplasm unclassifiable and Mast cell disease [7]. Indeed, the successive discovery of the gain-of-function *V617F* mutation in *JAK2* in 2005 [8] and the *JAK2* exon 12 mutation in 2007 [9], respectively, allowed a major advance in molecular characterization of these *BCR::ABL1*-negative MPN. The detection of *JAK2* mutations or other clonal markers became a major criterion of diagnosis in MPN classification and emphasized the clonal nature of these disorders. *JAK2V617F* mutation can drive PV, ET and PMF through constitutive activation of the receptors for erythropoietin (EPO-R), thrombopoietin (MPL) and granulocyte-colony stimulating factor (G-CSF-R) [10]. Over the years, other main driver mutations were discovered: MyeloProliferative Leukemia virus oncogene (*MPL*) exon 10 in 2006 [11] and Calreticulin (*CALR*) exon 9 in 2013 [12,13], leading to constitutive activation of JAK/STAT, PI3/AKT and MAPK/ERK signaling pathways [14], and are usually mutually exclusive. For the *MPL* mutations, the most prevalent gain-of-function mutations are *W515L* and *W515K* [15], inducing constitutive activation of JAK/STAT(5)(1) and PI3K/mTOR signaling pathways implicated in survival and proliferation of megakaryocytes (MKs) [16]. Two principal types of *CALR* mutants are observed: *CALR del52* (type 1), more frequent, and *CALR ins5* (type 2) [12,13], which display various phenotypes and prognosis. They are exclusive and more frequent after *JAK2* mutations. *CALR* mutants acquire new properties and bind to MPL, which lead to constitutive activation of MPL and corresponding signaling pathways [15,17].

In the 2016 revision classification, the detection of *JAK2*, *MPL* and *CALR* mutations belongs to major diagnosis criteria in PV, ET and PMF and are considered driver events. On the one hand, molecular analysis takes a preponderant place in the *BCR::ABL1*-negative MPN diagnosis and on the other hand, the bone marrow (BM) biopsy becomes nearly mandatory as a part of *BCR::ABL1*-negative MPN diagnosis to avoid missing out. Notably, the 2016 WHO classification removes mastocytosis entity, improves the criteria of the accelerated phase (AP) in CML and splits the PMF into two distinct entities because of their different prognosis: prefibrotic/early stage (pre-PMF) and overt fibrotic stage (PMF) on the basis cytomorphology analysis of BM [18].

The last 2022 classification now contains eight entities with the introduction of Juvenile MyeloMonocytic Leukemia (JMML). A few modifications within the entities are published and presented below in different sections. The fifth edition particularly insists on the idea of clonality with both the definition of Clonal Hematopoiesis and Clonal Hematopoiesis of Indeterminate Potential (CHIP). The latter is defined as the presence of a population derived from a multipotent stem/progenitor cell-harboring somatic mutations in genes associated with other myeloid malignancies (Acute Leukemia (AL) and Myelodysplastic Syndrome (MDS)) [19]. For each somatic mutation detected in the peripheral blood (PB) and/or in the BM, a significance threshold value for the Variant Allele Fraction (VAF) of 2% is proposed in order to harmonize practice between laboratories of molecular biology. Moreover, in the section on MPN, preponderant additional mutations others than *JAK2*, *MPL* and *CALR* are detailed. They provide proof of clonality and mainly deal with their prognostic impact in PMF. This classification insists on the importance of the association of quantitative data from the hemogram, cytological analysis of BM as well as molecular data to increase the diagnostic specificity and establish a better prognostic approach [19].

In this review, we will focus on the morphological appearance of PB and BM in the main *BCR::ABL1*-negative (PV, ET and PMF) and positive (CML) MPN, and we will insist on the association of specific cytological aspects and molecular data. These molecular data are more and more abundant and essential in the diagnostic establishment. It further permits a better knowledge of molecular mechanisms, therapeutic targets and disease prognosis.

## 2. *BCR::ABL1*-Negative MPN

At a physiopathological level, *BCR::ABL1*-negative MPN (including PV, ET and PMF) are mainly characterized by a medullar hyperplasia with repercussions on one or more blood cells lines and by a trend to extramedullary hematopoiesis [20]. Conversely to AL, the myeloid maturation is still conserved, and no excess of blastic cells is found. Different hemogram perturbations point towards MPN, notably elevation of hematocrit (Hct), hemoglobin (Hb), red blood cells (RBC) count in PV, chronic thrombocytosis in ET, erythroleukocytosis [21], teardrop-shaped RBC in PMF, even thrombocytosis and leukocytosis in the initial phase of PMF. Within the same entity of *BCR::ABL1*-negative MPN, it still exists a real heterogeneity according to the disease stage but also the mutations found. Moreover, there is a continuum and thin borders among *BCR::ABL1*-negative MPN above all in *JAK2V617F*-positive neoplasms [15].

*BCR::ABL1*-negative MPN are rather older adult diseases with a median age at diagnosis at 65–67 years [1]. Nevertheless, *CALR*-mutated and “triple-negative” ET patients are diagnosed at a younger age (49 years) than other ET [22]. These diseases are more frequent in males than in females with a sex ratio comprised between 1.4–2.3, except ET in which a sex ratio is comprised between 0.5–0.7. Nowadays, PV is the most prevalent MPN with 9.2–30 per 100,000 [1] whereas PMF prevalence is the lowest (1.76–4.05 per 100,000) [23]. Various risk factors of MPN are described, such as autoimmune diseases, standard immunosuppressive therapies, and tobacco smoking. This last one shows an increased risk to develop PV but not ET [1]. Inflammation can play a role in the MPN progression principally in PMF [24,25].

### 2.1. Polycythemia Vera (PV)

Polycythemia vera (PV) is an MPN characterized by clonal erythrocytosis which can be composed of two phases: polycythemic phase and secondary myelofibrosis phase. PV diagnosis is defined thanks to the data of the hemogram, blood mass evaluation, BM biopsy and molecular biology.

In the case of polycythemia detection on blood count, it is necessary to distinguish relative erythrocytosis (increased Red Cell Mass (RCM) and decreased plasma volume, thus Total Blood Mass (TBM) unchanged), from absolute erythrocytosis (increased RCM and unchanged or increased plasma volume, thus elevated TBM) [20] and primitive (PV and congenital polycythemia with erythropoietin (EPO) receptor gene mutations) from secondary polycythemias (congenital origin with a more affine hemoglobin for dioxygen, or EPO hyperproduction by acquired causes (prolonged hypoxia, inappropriate EPO secretion)) [26].

According to WHO classifications, this diagnosis requires major and/or minor criteria, defined in Table 1. Compared to the 2008 WHO classification, the 2016 revision reduced the threshold values of Hb and Hct levels to avoid missing masked PV (mPV) and promoted BM morphology from a minor to a major diagnostic criterion by recognizing the reproducibility of characteristic morphological features [27]. Moreover, the functional test (endogenous erythroid colony growth) is totally suppressed. PV diagnosis requires the presence either all three major criteria or the first two major criteria and the minor criterion, as described in Table 1 [18]. BM biopsy reminds facultative when the Hb level is superior to 18.5 g/dL (Hct > 55.5%) in men or 16.5 g/dL in women (Hct > 49.5%) and if the third major criterion and the minor criterion are present [28]. The BM biopsy also allows the diagnosis of possible initial fibrosis based on reticular fibrosis quantification. The only change indicated in the last WHO classification is the removal of the isotypic measurement of RCM as a diagnostic criterion [19].

In the polycythemic phase, neutrophilia and rarely basophilia may be present on PB with occasional immature granulocytes but circulating blasts are usually not detected. The BM aspiration shows a trilineage hyperplasia (panmyelosis) with especially erythroid hyperplasia but without dyserythropoiesis. There is also slight granulocytic hyperplasia with no dysgranulopoiesis signs. At low magnification, we observe an increase in pleomorphic mature MKs with atypical features such as hyperlobulation and/or clusters formation as in Figure 1 [20,29]. The size of cells is very heterogenous without abnormal maturation. Pearl’s staining does not reveal any iron deposits.

Only erythrocytosis permits us to distinguish PV from other MPN; nevertheless, it is difficult to objectify this characteristic. To be sure that there is an absolute erythrocytosis, RCM must be above 125% [20]. As mentioned above, this criterion has been removed in 2022 WHO classification [19]. However, this characteristic is particularly interesting in the definition of the mPV, which is a sub-entity of PV able to mimic an ET. In mPV, also called “pre-polycythemic phase”, patients present obvious thrombocytosis and not really elevated Hb/Hct with lower EPO levels than in ET [30]. The examination of BM will be very useful to discriminate with ET. This particular entity, positive for *JAK2V617F* mutation, mimics an ET presentation and explains the reason why the 2016 WHO classification reduced the threshold values of Hb and Hct. In daily practice, complex cases or atypical presentations exist and require a multidisciplinary approach with a meticulous analysis of megakaryocytic lineage on both blood and medullar smears. The absence of platelets dystrophy nor large or giant size MKs in BM allows us to discuss a diagnosis of mPV rather than ET [18,29]. Reticulin staining is normal in 80% of cases but increased reticulin or mild to moderate collagen fibrosis can be observed, depending on the stage of disease [31]. At the diagnosis, the discovery of fibrosis grade 1 has a negative prognostic impact with the rapid evolution of the disease [32,33]. Therefore, BM biopsy should be realized whatever the levels of Hb, Hct and EPO for establishing the initial diagnosis.

Transformation into myelofibrosis can occur in 4.9–6% of PV at 10 years [33,34]. This stage is generally associated with cytopenia, extra-medullary hematopoiesis and hypersplenism. On the hemogram, this phase is characterized by a decreased RBC count, the presence of poikilocytosis with teardrop-shaped RBC and leukoerythroblastosis with neutrophilic on the blood smear (≥13 × 10^9^/L) [35,36]. The discovery of circulating blasts (≥10%) in PB indicates a transformation in AP. In MPN, the cytologic follow up is important to screen if indirect signs of myelofibrosis appear to inform clinicians of disease evolution as shown in Figure 2.

The BM biopsy shows lower cellularity: erythropoiesis and granulopoiesis are decreased with possible fibrosis of grade 2–3. Dystrophic megakaryocytic clusters with hyperchromatic and very dysmorphic nuclei (hypolobulation) are prominent and an elevation of blast count is observed [33]. Osteosclerosis and the transformation into MDS or AL (more than 20% blast count in PB or BM) are around 10% [20].

The molecular analysis finds a gain of function *JAK2V617F* mutation in exon 14 in more than 95% of PV cases in a homozygous state after mitotic recombination but it is not specific for this entity and is found also in 50% of patients with ET (heterozygous state), PMF (homozygous after mitotic recombination) [10] or Refractory Anemia with Ring Sideroblasts and Thrombocytosis (RARS-T) [37] and at a lower frequency in other myeloid neoplasms (in 8% of Chronic MyeloMonocytic Leukemia (CMML) for example) [38,39]. *JAK2V617F* mutation have been reported in rare CML patients with *BCR::ABL1* fusion gene, in independent clones probably [40,41,42,43,44]. Studies have proved that the phenotype of these different MPN could be modified by the allelic burden of the *JAK2V617F* mutation. An increased *JAK2V617F* allelic burden (>50%) or homozygous mutation [15] correlates with an increase in Hb and leukocytes count and a lower platelet count [45,46]. It is associated with an increased risk of thrombosis and fibrotic evolution [20,47,48]. In mPV, *JAK2V617F* allele burden is lower with less elevated Hb level and leukocytes count than PV [30,31]. *JAK2* exon 12 mutation is present in approximately 3% of PV cases and correlates with variable leukocytosis [49] and lower platelet count than *JAK2V617F* mutated cases but with higher Hb level and a subnormal serum EPO concentration. Indeed, like the *JAK2V617F* mutation in exon 14, the exon 12 mutation also induces erythropoietin hypersensitivity [9]. In BM, erythroid preponderance is observed [45] and associated with an increase in MKs with variable nuclear lobulation (only one lobe or hyperlobulation) and a large spectrum of sizes but notably a predominance of small MKs. Cluster formation is also possible but rare and subtle compared to *JAK2V617F* mutated cases. The reticulinic fibrosis is normal or discreetly majored. Nevertheless, nearly half of *JAK2* exon 12 mutated PV does not show megakaryocytic cytomorphologic abnormalities [14]. Both mutations have similar prognosis. Finally, no *MPL* mutation is observed in PV and only very rare *CALR* mutation type I can be observed [50]. In these two described cases, erythropoiesis is elevated with irregular MKs but without clustering [33].

In laboratory practice, there are sometimes complex diagnoses when both cytology and molecular analysis (VAF < 1%) [45] are negative but with an evocative clinic for PV. With the study of karyotypes (abnormal in 20% of patients) [51], the development of both whole-genome analysis (CGH microarrays) and myeloid Next-Generation Sequencing (NGS) technologies, the molecular landscape of MPN becomes considerably more complex with the discovery of numerous acquired genetic abnormalities. These abnormalities have an impact on the evolution of the diseases and permit us to explain the heterogeneity of MPN phenotypes. In addition to the *JAK2* mutations, around 50% of patients with PV have additional mutations at diagnosis time and their presence is associated with a high risk of progression [31,52]. As shown in Table 2, the most frequent mutations encountered are *TET2* (Ten-Eleven Translocation oncogene family member 2), *DNMT3A* (DNA cytosine MethylTransferase 3A)*, ASXL1* (Additional SeX combs-Like 1), *SF3B1* (RNA Splicing Factor 3B, subunit1) and *LNK* (lymphocyte adaptor protein (*SH2B3*), respectively implicated in DNA methylation, histone modification, mRNA splicing and negative regulation of hematopoietic growth factors signaling [51,52]. Some mutations have a prognostic impact as published for *SRSF2* (Serine/arginine-Rich Splicing Factor 2), *IDH2* (Isocitrate DeHydrogenase 1 and 2), *RUNX1* (Runx-related transcription factor 1) and *U2AF1* (U2 Small Nuclear RNA Auxiliary Factor 1). They are considered adverse mutations for overall, leukemia-free or myelofibrosis-free survival [31,51]. *SRSF2* and *RUNX1* mutations are more frequent in blast phase MPN [53].

The heterogeneity of the MPN evolution is not only explained by the type of mutated genes but also by their chronology of appearance and their allelic load. Regarding the order of appearance, an additional mutation can appear prior to or after a driver mutation and modify the phenotype as demonstrated with *TET2* and *DNMT3A* in *JAK2*-positive MPN [10,57]. For example, if *TET2* mutation is the first event in HSC, an ET phenotype will be induced with the expansion of single mutant cells in the HSC population limiting excess production of MKs and erythrocytes until the second hit with *JAK2V617F* mutation. In the case of “*JAK2*-first”, a PV phenotype with excessive production of RBC and an expansion of double mutant cells in the HSC population is observed. Comparably to *TET2*, if *DMT3A* mutation is acquired before *JAK2V617F*, the MPN patients more commonly develop an ET phenotype whereas a PV phenotype will be observed in the case of “*JAK2*-first”. Therefore, the mutational order affects both self-renewal of mutants in the HSC population and the proliferation driving to ET or PV phenotype [58].

The different PV diagnostic approaches are summarized in Figure 3.

### 2.2. Essential Thrombocythemia (ET)

Essential thrombocythemia (ET) is an MPN characterized by clonal thrombocytosis. Following the 2016 WHO classification, ET requires all four major criteria or the first three major and minor criteria [18], as described in Table 3. In the 2022 WHO classification, nothing was changed in diagnostic criteria [19].

Compared to the 2008 WHO classification, the criteria are not really changed, except the definition of *MPL* and *CALR* mutations, which have been described in 2006 and 2013, respectively [11,12,13]. With the introduction of this molecular biology criterion, the criterion of reactional thrombocytosis moves to a minor criterion. However, the presence of reactional thrombocytosis does not formally exclude the hypothesis of underlying ET hence the importance of detecting the three driver mutations (*JAK2*, *MPL* and *CALR*) in persistent thrombocytosis. ET diagnosis is especially difficult because an isolated platelet elevation superior to 450 × 10^9^/L can be found in many situations, such as post-splenectomia, post chirurgical or obstetrical stress, iron deficiency, inflammatory status, corticotherapy or cancers [59], and in contrast to PV, about 10% of ET patients are triple-negative genotypes [22]. Thus, ET remains an exclusion diagnosis. Moreover, most of the other myeloid neoplasms can mimic ET: CML with thrombocytosis and especially the pre-PMF. In the first case, the detection of *BCR::ABL1* permits the distinction whereas in the second case, molecular analysis will not permit it. In this case, some cytologic clues can help to distinguish the two entities: in ET there is typically no anemia nor abnormalities in RBC indices and there is no teardrop-shaped RBC on blood smear [48]. The morphologic analysis of BM will be determinant as seen below [60,61].

In ET, an elevated platelet count will be the main abnormality in routine hemogram. Sometimes, a moderate leukocytosis composed of neutrophils can be associated. Often, platelet can present morphologic abnormalities, observed on the blood smear: macroplatelets, giant platelets, strange shapes, pseudopods, agranular platelets or circulating micromegakaryocytes in post-ET PMF [62]. BM smears show normo- or hypercellularity with an important megakaryocytic hyperplasia [20]. Erythroblastic and granulocytic lineages are correctly represented [33].

On the myelogram, 20 to 49% of the total MKs are large to giant [63], with abundant, mature cytoplasm associated with hyperlobulated “staghorn-like” nuclei [33]. They are dispersed into the BM sometimes in loose clusters formation [48] as shown in Figure 4.

BM cytomorphology is crucial to differential diagnosis. In reactive thrombocytosis, MKs are increased but do not present morphological abnormalities, they are mature with sometimes a pleomorphism [64]. In CML, MKs are typically reduced in size with normal chromatin pattern [29] and in MDS/MPN with ring sideroblasts, the MKs have different sizes from small to giant with the Pearl staining highlighting the ring sideroblasts [48]. An increase in erythroblastic and granulocytic lineages should lead to a mPV in case of *JAK2V617F* mutation associated [33]. The presence of dense clusters of MKs and atypical MKs will be in favor of pre-PMF. Reticulin staining is usually normal at diagnosis in ET; grade 1 fibrosis can be observed only in less than 5% of cases [65]. This key point is very important in differential diagnosis with pre-PMF. Nowadays, standardized machine learning tools are developed and permit the improvement of the definition of reticulin fibrosis. This aspect is interesting to harmonize the quantitation of this fibrosis in different laboratories [66]. Indeed, ET and pre-PMF do not share the same risk of progression towards myelofibrosis (at 15 years, 9% versus 17% in ET and pre-PMF, respectively) and leukemic transformation (at 15 years, 2% versus 12% in ET and pre-PMF, respectively) [45,67]. Rare patients (less than 5% at 10 years) display an evolute form with indirect signs of myelofibrosis on the blood smear (poikilocytosis, teardrop-shaped RBC, leukoerythroblastosis), or circulating blasts (≥20%) in the case of transformation in AL [68]. In case of indirect myelofibrosis signs on the blood smear, the BM aspiration and BM biopsy can eventually show grade 2–3 fibrosis features [33].

On the molecular aspect, *JAK2V617F* mutant is present in approximately 50% of cases [14] with usually a VAF less than 25% whereas *JAK2* exon 12 mutation is not observed [33]. ET with a high *JAK2V617F* allelic burden is more susceptible to evolving into PV or PMF. Moreover, a “PV-like” phenotype is described in *JAK2V617F*-positive ET and not in *JAK2V617F*-negative ET [20]. Thrombosis is more frequent in *JAK2V617F*-positive ET patients [69]. *MPL* mutations are present in 3–8% of ET [22] and are associated with an increase in both platelet count and serum EPO level, and with both lower Hb level and marrow cellularity than in *JAK2V617F*-positive ET patients [54,70,71]. In a few patients, somatic *MPLS505N* mutations had been reported although this mutation was initially associated with inherited thrombocytosis [70]. *MPL*-positive ET is more associated with an evolution towards PMF [72]. *CALR* mutants are present in 30% of ET and are more frequent in young patients with less thrombotic risk [73]. Moreover, *CALR* mutations are found in 67–71% of ET that are not mutated for *JAK2* and *MPL* [12,13]. They are exclusive and more frequent after *JAK2* mutation. Comparatively, with *JAK2V617F*-mutated patients, *CALR*-mutated ET are associated with higher platelet count, lower Hb level and leukocyte count [48]. Type 1 *CALR* mutation is more frequent in ET (50%) [74,75] and shows a higher risk to evolve towards myelofibrosis [15,48,76]. Type 2 *CALR* mutation is observed in young patients and is associated with higher platelet count but with a lower incidence of thrombosis at diagnosis and less myelofibrotic transformation.

On the morphologic axis, in *JAK2V617F* and *CALR*-mutated ET, the size of MKs is twice as large as the *MPL*-positive ET or triple-negative ET reported by experimented cytologists. In triple-negative ET (around 10% of ET), the platelet count is very elevated with specific morphologic abnormalities on MKs and is an indolent disease with less vascular events [22]. Therefore, in these cases, the myeloid NGS is very interesting to prove clonal hematopoiesis (with VAF ≥ 2%) above all in cases of mutations of genes implicated in epigenetic regulation (*TET2*, *DNMT3A*, *ASXL1*, *EZH2* and *IDH1/2*), in mRNA splicing (*SF3B1*, *SRSF2* and *U2AF1*) and in the regulation of cytokine signaling (*CBL* (casitas B lineage lymphoma protooncogene)) [10]. In PV, 53% of ET patients have one or more sequence variants/mutations revealed by NGS studies [52]. The *CALR*-positive MPN have a less complex molecular landscape than *JAK2*-positive MPN [77]. When *CALR* mutation is associated with an additional mutation like *ASXL1*, median Hb level is lower than only *CALR*-mutated ET. In *JAK2V617F*-positive ET, when patients are *TET2* or *DNMT3* mutated before the driver mutation, the ET phenotype is preponderant [57,58]. As in PV, the *SRSF2* mutation seems to be associated with a worse prognosis in ET, as well as the *SF3B1*, *U2AF1* and *TP53* mutations [51,52,78] The monitoring of *CALR* allele burden seems to be interesting to predict evolution [74] as for additional mutations, the study of the rate of the allele burden seems to be interesting for the leukemic transformation process [79].

### 2.3. Primitive Myelofibrosis (PMF)

PMF is the least common and most aggressive MPN manifested by BM fibrosis, extramedullary hematopoiesis (mainly in the spleen and/or liver) and inappropriate production of cytokines [23,80]. Myelofibrosis can be primitive or secondary to others, such as MPN or MDS, infection, inflammatory or autoimmune diseases, metabolic diseases and neoplastic disorders [60]. PMF evolves in two phases: the prefibrotic PMF/early PMF (pre-PMF) corresponding from 30 to 50% of PMF cases with no significant increase in reticulin or collagen (grade 0 or 1) and the fibrotic phase (overt PMF) with reticulin or collagen fibrosis of grade 2 or 3 [18]. Compared to the 2008 version, the 2016 WHO classification introduces the pre-PMF phase. The diagnosis of this phase requires three major criteria and at least one minor criterion confirmed in two consecutive determinations, as described below in Table 4 [18]. Molecular criteria take a preponderant place because besides the three well-known driver mutations, it is more and more necessary to search clonality markers as *ASXL1*, *DNMT3A, EZH2*, *TET2*, *IDH1/2*, *SRSF2* and *SF3B1* because they are more frequent in PMF and in advanced MPN. The 2022 WHO classification insists on this molecular aspect because the presence of additional mutations (*ASXL1*, *EZH2*, *IDH1/2*, *SRSF2* and *U2AF1)* are associated with poor outcomes and called “High-Risk Mutations” (HRM) [19,47].

At the pre-PMF stage, hemogram usually shows anemia, moderate leukocytosis (>11 × 10^9^/L) and/or thrombocytosis (25% of patients) with macroplatelets and giant platelets on blood smear, as shown in Figure 5.

Due to this thrombocytosis, it is necessary to differentiate pre-PMF from ET based on morphologic findings in the BM biopsy as detailed in WHO 2022 classification. Pre-PMF demonstrates hypercellular BM with granulocytic proliferation with often erythropoiesis hypoplasia without dysplasia [15] and without immature cells excess, nor maturation blockade. Megakaryocytic abnormalities which are better appreciated by the BM biopsy are the key factors for the diagnosis with analysis of topographical distribution in BM. There are numerous MKs with variable sizes (small to high), pleomorphic with aberrant nuclei–cytoplasmic ratio, hyperchromatic nuclei with variable lobulation (bulbous, cloud-like or balloon-shaped nuclei), and the presence of naked megakaryocytic nuclei, as shown in Figure 6 [29,63]. These are often arranged in dense clusters adjacent to BM vascular sinuses and bone trabeculae [48]. With the new machine learning approach, it is possible to precisely distinguish the morphological features of MKs and MPN with high accuracy [81].

Compared with other MPN, in pre-PMF, MKs are more atypical and represent a strong argument to diagnose pre-PMF and not ET [61]. This cytological distinction is very important because pre-PMF will evolve more often and/or rapidly towards overt PMF than ET [67,82]. Lymphoid reactional nodules are present in about a quarter of cases [83]. BM aspiration shows the same cytologic abnormalities and it is complementary to the biopsy. At this stage, the BM biopsy does not highlight significative fibrosis, a maximum of grade 1 [84]. Cytomorphological aspects of PMF are summarized in Table 5.

In the case of overt PMF, major criteria have not changed in 2016 compared to 2008 but details are noticed about grades of fibrosis [18]. In contrast, there is an additional minor criteria which is the presence of leukocytosis ≥ 11 × 10^9^/L [28]. At this stage, hemogram usually shows anemia and a lower platelet count. It is also possible to observe leukopenia. On blood smear, there is often an erythroleukocytosis [18] and an anisopoikilocytosis with teardrop-shaped RBC which are characteristic features but not specific. Circulating immature cells and MKs of different shapes (high size, micromegakaryocytes, nude nuclei or classical MKs) may also be present. There is an increase in the number of circulating CD34+ cells in the later stages in association with extramedullary hematopoiesis, usually more than 15 per microliter [85]. Monocytosis (>1 × 10^9^/L) is a sign of evolution in PMF [86]. To distinguish this evolution from CMML, it is important to be attentive to cytological indirect signs of the PB and BM biopsy, evoking overt PMF. Moreover, the molecular analysis should be performed for the research of MPN driver mutations to avoid a misdiagnosed CMML [19,87]. The main difference with the prefibrotic stage is the apparition of reticulinic and/or collagenic fibrosis > grade 1 [18]. Due to fibrosis, the BM aspirate is typically hypocellular and often unhelpful. It shows some dystrophic MKs and rare erythroblastic islets. BM biopsy shows heterogenic cellularity with the coexistence of both conserved hematopoietic (in vascular sinuses) and fibrotic clusters. There is also the presence of significant megakaryocytic dystrophia grouped together in clusters of variable size [83]. A redistribution of fat cells along the bone trabecules is observed in BM PMF [33]. Lymphocyte infiltration and reactional plasmacytosis are frequent. Immature CD34+ cells remain inferior to 10% but an evolution in AL is possible and happens for 5–30% of patients [83]. Due to phenotypic mimicry of a lot of diseases (malignant or not) [23], PMF diagnosis is difficult and BM biopsy has a major role in its diagnosis [20]. Guidelines were established to define semiquantitative BM fibrosis grading system or semiquantitative grading of collagen and osteosclerosis [88] in order to use reproductible tools (for example: machine learning approach to quantify reticulin fibrosis like continuous indexing of fibrosis) [66]. In the 2022 WHO classification, experts insist on this aspect for monitoring BM fibrosis in case of treatment by JAK1/2 inhibitors [19]. In myelofibrosis following pre-PMF, PV or ET, the level of reticulin fibrosis is the main component, followed by an increase in the number of collagen fibers [60]. Moreover, some links are now highlighted between cytomorphologic and molecular features.

In PMF (early or advanced stages), mutations of *JAK2V617F*, *JAK2* exon 12, *CALR* and *MPL* are found in, respectively, 50%, 2–3%, 25–30% and 13.5% of cases [23,33]. Moreover, *CALR* mutations are found in 56–88% of PMF that are not mutated for *JAK2* and *MPL* [12,13]. About 12% of PMF are triple-negative and correspond to difficult diagnosis [22]. *JAK2V617F*-positive PMF cases show a higher Hb level, white blood cell and platelet counts associated with an elevated allele burden. In these patients, an elevated CRP was observed, with an increased IL-1 receptor antagonist (RA), IL-10 and IL-2R. Elevated CRP was described as associated with a bad survival [89]. Due to TPO hypersensitivity, an *MPL* mutation, associated or not to *CALR* mutation, leads only to an increase in platelet count and megakaryocytopoiesis. *MPLW515L*-positive PMF present a more severe anemia [90]. Type 1 *CALR* mutation is associated with a higher platelet count and less incidence of anemia and leukocytosis [48] thus survival is favorable [53,76]. Among *CALR* mutations, *CALR del52* is more frequent in PMF (70%) than *CALR ins5* (13%) [74,75,91]. As it was shown that the rate of *JAK2* allele burden could modify the phenotype of MPN between ET, PV and PMF, the rate of *CALR* allele burden usually found in the initial clone, is more elevated in PMF than in ET [74]. In triple-negative myelofibrosis the prognostic is pejorative with a significantly higher risk of leukemic evolution [22]. A total of 40% of PMF patients have an abnormal karyotype with commonly 20q−, 13q−, +8, +9, 12p−, inv(3), −5/5q− and 11q23 rearrangements and abnormalities in chromosomes 1 and 7 [56,92]. From the 2016 WHO classification and with the development of myeloid NGS, molecular analysis takes a preponderant place. In PMF (primary or secondary), more than 80% of patients have genomic variants in myeloid genes: *ASXL1*, *TET2*, *SRSF2*, *EZH2* and *IDH1/2* that are the most involved genes and *TP53*, *EZH2*, *SRSF2* and *U2AF1* mutations are associated with adverse prognostic likewise for *IDH1/2*, *CBL* and *K/NRAS* [56,93]. Among the numerous studies published, the number of mutations can influence prognosis. For example, *EZH2*-mutated PMF patients have a higher leukocytes count with blastic cells and more important splenomegaly [53]. Stable MPN have either no additional mutations or one additional mutation at diagnosis while unstable MPN have around 4 additional mutations reflecting genetic instability during disease progression [77,79]. Mutations in *DNMT3A*, *IDH1/2*, *TP53* and *SRSF2* genes are more frequent in MPN evolving into AML [56]. *DNMT3A* and *TP53* mutations are more frequent in post PV/ET AML than in post-PMF AML [77] and mutations in *U2AF1* are more associated with anemia and short survival [23]. *TP53* mutations are considered as a late event because they are observed at the leukemic transformation in post-MPN AL although they may be present at diagnosis at a low frequency [79,93]. International prognostic scores involving additional mutations were developed for PMF, PV and ET to identify patients with the risk of progression. In PMF, additional mutations play a central role in the development of prognostic scores: IPSS, DIPSS, DIPSS-Plus, MIPSS70, MIPSS70+ version 2, GIPSS and MYSEC-PM for secondary myelofibrosis to categorize patients for the risk of evolution. GIPPS is exclusively based on genetic and cytogenetic data [51,53,67,94,95]. Therefore, in clinical practice, NGS is recommended to improve risk stratification even if it is not mandatory to establish the optimal treatment [78]. Mutation of *ASXL1* was initially associated with an unfavorable prognosis but in a recent French study, the impact on prognosis could be different if *ASXL1* is alone or associated with *TP53* or high-risk mutations [93,96]. Thus, it becomes more and more complex to define the prognostic of patients.

## 3. *BCR::ABL1*-Positive MPN: Chronic Myeloid Leukemia (CML)

Chronic Myeloid Leukemia (CML) is a rare MPN with an incidence of 1 case per 100,000 people [97]. Men are more affected than women with a sex ratio of 1.4 and a median age at diagnosis of 61 and 62 years for men and women, respectively [98]. Before the era of the development of specific Tyrosine Kinase Inhibitors (TKIs), untreated CML was initially composed of three phases: chronic phase (CP), accelerated phase (AP) and blast phase (BP) with AP noticeable. In case of untreated CML or with a bad response to TKIs, CML can evolve to BP which has a poor prognosis (median survival of 12 months). AP has an intermediate prognosis which is improved by using TKIs [3]. The criteria of different stages are reported in WHO classifications (used by cytopathologists) and in the European Leukemia Net (ELN) classification (used in therapeutic studies with a threshold of the blast at 30% to define BP-CML) [99]. Since 2001, CML is defined by the presence of Philadelphia Chromosome or *BCR::ABL1* fusion gene. AP and BP were clearly defined to improve the practice [100]. Over the WHO classifications, these criteria for AP and BP were optimized as described in Table 6. With the success of TKIs therapy (the 10-year overall survival rate for CML is now 80–90%) and its conscientious monitoring, AP decreased and in the fifth edition of the 2022 WHO classification, the designation of AP has completely disappeared in favor of an emphasis on high-risk features associated with BP progression and resistance to TKIs [19].

Most CML patients are diagnosed with CP which is frequently asymptomatic. Rarely are patients diagnosed directly with lymphoid or myeloid BP without CP. The diagnosis of CML is usually based on important leukocytosis (12–1000 × 10^9^/L, median 80 × 10^9^/L) with excessive granulocytosis with metamyelocytes, myelocytes, promyelocytes and sometimes the presence of circulating myeloblasts (<2%). Eosinophilia and basophilia are generally associated. Nowadays, the diagnosis is earlier with low leukocytosis and only a few immature granulocytes in PB. Platelet count is normal or rarely elevated above 1000 × 10^9^/L [101]. Thrombocytopenia is often associated with AP or BP. On blood smear, granulocytic dysplasia is absent, and the proportion of monocytes is usually < 3% of leukocytes. The presence of eosinophils and basophils helps for a differential diagnosis such as hyperleukocytosis secondary to post-aplasia regeneration, agranulocytosis, major hemolysis, infection, or cancer metastases (especially in low white blood cells forms).

In the majority of cases, CML is diagnosed both on the hemogram data and on the detection of the Ph chromosome by cytogenetic and/or *BCR::ABL1* gene fusion by molecular genetic techniques. However, the BM aspiration is important to confirm the phase of the disease with the quantification of blastic cells [18]. In CP CML, BM analysis shows trilineage hyperplasia with predominant granulocytic hyperplasia with eosinophilia and basophilia without significant dysplasia, as shown in Figure 7a. The proportion of erythroid lineage is usually decreased. In contrast, MKs are increased in 40–50% of cases with small size and hyposegmented nuclei, referred to as “dwarf” MKs with intermediate N/C ratio, normal chromatin pattern and cytoplasmic differentiation as shown in Figure 7b [29] but they are not microMKs as seen in MDS. Moreover, pseudo-Gaucher cells are usually observed in BM due to hypercatabolism. Commonly, blasts are less than 5% of the BM cells.

In previous WHO classifications, AP CML was diagnosed by an excess of blasts (10–19%). An excess of microMKs is also observed, suggesting additional cytogenetic abnormalities (ACA) and an adverse outcome. To prove fibrosis, a BM biopsy is necessary for a significant reticulin and/or collagen fibrosis. In BP, the criteria are the presence of 20% or more of blasts in the PB or BM or the presence of an extramedullary proliferation of blasts, as detailed in Table 6. In most BP, blasts are of a myeloid origin and may include neutrophilic, monocytic, megakaryocytic, basophilic, eosinophilic or erythroid blasts or any combinations. In 20–30% of BP, the blasts are lymphoblasts (usually B-cell origin) [100]. BM biopsy is not recommended to diagnose CML but it is necessary if the presentation is atypical or if cellular aspirate cannot be obtained to diagnose CML. In effect, in approximately 1% of CML, an isolated thrombocytosis is only observed [102]. Therefore, in case of persistent thrombocytosis without leukocytosis and negative driver mutations, *BCR::ABL1* will be researched to avoid missing atypical presentation of CML. Although, classical driver mutations are mutually exclusive, rare cases of CML with *JAK2* or *CALR* mutations have been observed [44,102,103]. The temporal sequence of these mutations is variable suggesting that the alterations might reside in the same or distinct clones [40,42,43]. Sometimes CML with thrombocytosis and leukocytosis mask a Ph-negative MPN which is only discovered after successful TKI therapy [104,105,106,107,108]. BM examination above all cytomorphological of MKs is very important in differential diagnosis in TE and pre-PMF or MDS/MPN (RARS-T) [109]. In case of persistent monocytosis with or without immature granulocytes in PB, signs of dysgranulopoiesis will be researched on blood smears to avoid a misdiagnosis of CMML.

The confirmation of the diagnosis is obtained by the identification of the classical balanced translocation t(9;22)(q34;q11) in karyotype in 95% of cases. In 5% of cases, it can also take the form of a complex rearrangement with the implication of other chromosomes (t(9;22;v)) or a cryptic abnormality where Ph is masked (for example in the case of insertion). Then, the confirmation of the diagnosis is based on highlighting of *BCR::ABL1* fusion gene either by fluorescent-in-situ-hybridization (FISH) and/or by reverse transcriptase PCR (RT-PCR). In every case, it conduces to the fusion of the 5′ end of *BCR* (22q11) gene with the 3′ end of *ABL1* (9q34) gene and the generation of an oncogene that leads to the expression of a chimeric constitutively active tyrosine kinase [3]. The site of the breakpoint may influence the phenotype of the disease. In 95% of patients, the breakpoint is located in exon 13 or 14 of *BCR* leading to a protein of 210 kDa (M-BCR) with e14a2 mostly observed. More rarely, it can occur in exon 19 (e19a2) with a BCR-ABL1 protein of 230 kDa (µ-BCR). This protein is more observed in patients with important neutrophilia. Patients with thrombocytosis at diagnosis can express p210 or p230 kDa proteins [18,110,111]. Occasionally in CML and especially in Acute Lymphoblastic Leukemia (ALL), a BCR-ABL1 protein of 190kDa (m-BCR) is found due to a breakpoint in exon 1 (e1a2). This protein is also observed in rare cases of CML with monocytosis as well as the p210kDa protein [112,113,114]. The research of *BCR::ABL1* is an important step for differential diagnosis in the case of persistent leukocytosis and/or thrombocytosis (after eliminating reactional events), notably, if there are few myeloid precursors. It will be important to think about LCN or MDS/MPN with neutrophilia and the molecular approach will focus on the research of other mutations in the genes *CSF3R* or *SETBP1*.

Clonal evolution is described as a multistep process associated with genomic instability which occurred either late in CP or early in ex-AP, before the accumulation of blasts. In 5–10% of cases [115] at diagnosis, we can observe ACA at the Ph chromosome with in particular the duplication of the Ph, the 17q isochromosome and a gain of chromosome 8. The trisomy of 19, 21 and 17 chromosomes, the loss of Y chromosome or monosomy 7, represent less than 10% of clonal evolutions [6]. The frequency of these ACA reaches 80% in case of BP [116]. The identification of these abnormalities represented key criteria for AP diagnosis. Despite the removal of AP in the fifth edition of the WHO classification in 2022, the presence of “high risk” ACA (+8, +Ph, i(17q), +19, −7/del(7q), 11q23(*KMT2A*) rearrangements, 3q26(*MECOM*) rearrangements, complex karyotype) remains an important feature for prognosis and risk of progression [99]. The presence of iso(17q), −7 and 3q26.2 rearrangement is associated with an adverse prognosis [117]. The impact of the ACA in cells without the Ph chromosome is not actually well defined in terms of prognosis [118]. Sometimes, the cytological analysis of PB and BM in the follow-up of the CML permits us to detect the appearance of under clones, in particular in the presence of numerous microMKs. In this case, a cytogenetic analysis will be interesting in order to search additional abnormalities as t(3;21)(q26;q22) involving the *RUNX1* and *MECOM* genes, objectifying an acceleration of the disease and requiring modified management by favoring a Hematopoietic Stem Cell Transplantation (HSCT). Moreover, as in the other MPN, somatic mutations are found in epigenetic regulators, *ASXL1*, *TET2, TET3, DNMT3A* or in transcription factor, *RUNX1* [6,119] which seem to be acquired in the Ph+ cells [120]. *RUNX1* is mutated in the blastic phase suggesting that this mutation could be implicated in the progression of the disease [119]. *ASXL1* mutation (within exon 12) is the most frequent somatic mutation in Ph+ clones and disappears with the effectiveness of TKIs treatment in 3 or 6 months [120]. However, the prognostic impact is not clearly defined.

In the case of resistance to TKIs treatment, molecular analysis is necessary to search mutations in *BCR::ABL1* kinase domain (KD), described as the main mechanism of resistance to TKIs, in association with monitoring of TKI plasma assay. These mutations in *BCR::ABL1* kinase domain (KD) are not found in one-third of resistant patients. Therefore, molecular analysis is necessary to explain the unknown implicated process. Recent studies showed that mutations in genes such as *RUNX1*, *ASXL1* and *IKZF1* deletion could be implicated in drug resistance [47]. Indeed, the detection of mutational profiles could be useful to detect high-risk and resistant patients in the coming years [121].

The different diagnostic approaches for CML, ET and pre-PMF are summarized in Figure 8.

## 4. Conclusions

MPN are a heterogenous subgroup of clonal diseases characterized by an abnormal proliferation of hematopoietic stem cells in the BM. For the last fifty years, the discovery of the fusion between the two genes *BCR* and *ABL1* and three driver genes (*JAK2*, *MPL* and *CALR)* permitted a better knowledge of MPN pathophysiology. The common point of these alterations is the activation of different signaling pathways implicated in a survival and proliferative advantage. Each molecular abnormality drives the clinical phenotype of MPN. *BCR::ABL1* is found in 95% of CML, in which leukocytosis with granulocytic precursors, eosinophilia and basophilia are the most preponderant abnormalities on the hemogram. The mutations in the *JAK2*, *MPL* and *CALR* genes are responsible for myeloproliferation by directly or indirectly deregulating JAK2 signaling resulting in two molecular entities: *JAK2V617F* mutated MPN where the same mutation can lead to three distinct phenotypes (ET, PV and PMF) depending on its allelic load and, *MPL/CALR* mutated MPN including ET and PMF. These driver mutations are mutually exclusive and are present in 90% of PV, ET and PMF, explaining their inclusion in the 2016 WHO classification as major diagnostic criteria. Over the years, the characterization of MPN was improved on the phenotypic aspect with more performant tools for the detection of cell abnormalities in the PB. Moreover, a better harmonization of practices in BM cytology allows us to place the BM biopsy in the major diagnostic criteria of *BCR::ABL1*-negative MPN. Indeed, an attentive cytomorphologic examination of BM (cellularity, ratio erythropoiesis/granulopoiesis, specific features of MKs, BM fiber content) is essential in the discussion about differential diagnosis and in some difficult or atypical *BCR::ABL1*-negative MPN cases, above all in “triple negative” MPN. With the development of molecular biology platforms, the molecular landscape of MPN is more and more complex with the discovery of numerous somatic mutations and variants of genes implicated in different cellular functions. Around 50% of MPN are already mutated at the diagnosis and the mutation frequency increases with the evolution. Molecular heterogeneity (type of mutated genes but also their chronology of appearance and their allelic load) as well as the bone marrow microenvironment could be at the origin of the prognostic heterogeneity of MPN in terms of progression towards myelofibrosis and leukemic transformation. Notably, the most frequent mutations in PMF are *ASXL1*, *DNMT3A, EZH2*, *TET2*, *IDH1/2*, *SRSF2* and *SF3B1* and some of them are now well known, called HRM *(ASXL1*, *EZH2*, *IDH1/2*, *SRSF2*, *U2AF1*), and are included in prognostic scores.

MPN are increasingly well-defined on the phenotypic and molecular aspects which have been permitted to establish powerful prognostic scores. In the future, it will be easier to personalize the treatment with targeted therapies and the molecular follow-up could be performed in the same way as for the CML.

## Figures and Tables

**Figure 1 cells-12-00946-f001:**
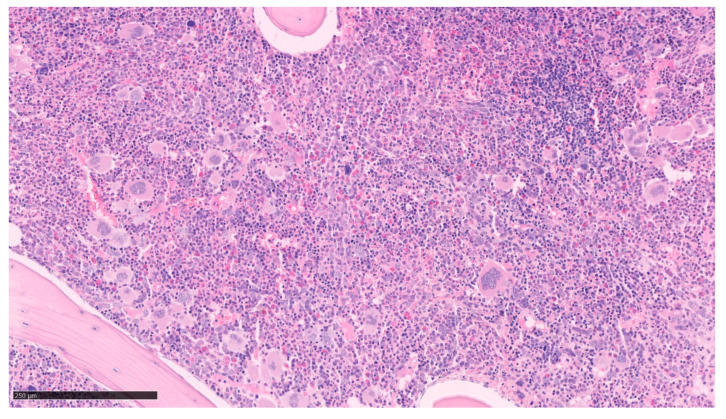
Bone marrow biopsy of Polycythemia Vera. Panmyelosis with an increase in pleomorphic mature megakaryocytes. (optical microscopy, May Grünwald Giemsa (MGG), low magnification ×15).

**Figure 2 cells-12-00946-f002:**
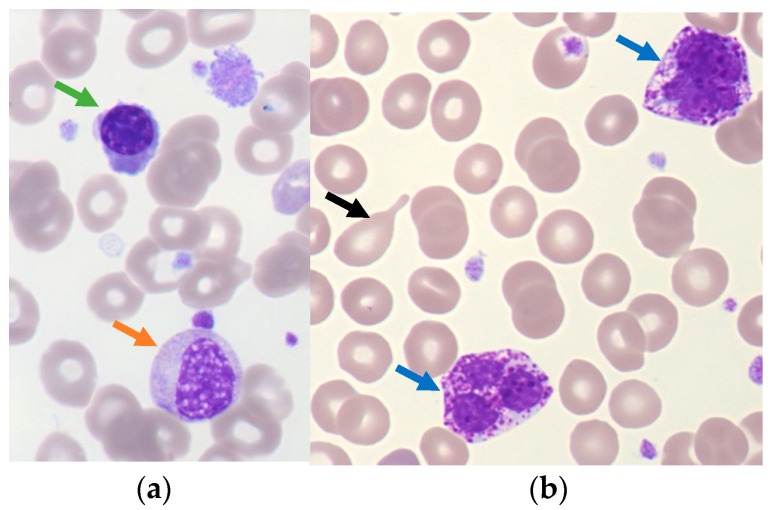
Indirect signs of myelofibrosis on blood smear. (**a**) Leukoerythroblastosis with a polychromatophilic erythroblast cell (green arrow) and immature granulocytic cell (orange arrow); (**b**) Teardrop-shaped RBC (black arrow) and basophilic cell (blue arrow). (Optical microscopy, May Grünwald Giemsa, ×50 magnification).

**Figure 3 cells-12-00946-f003:**
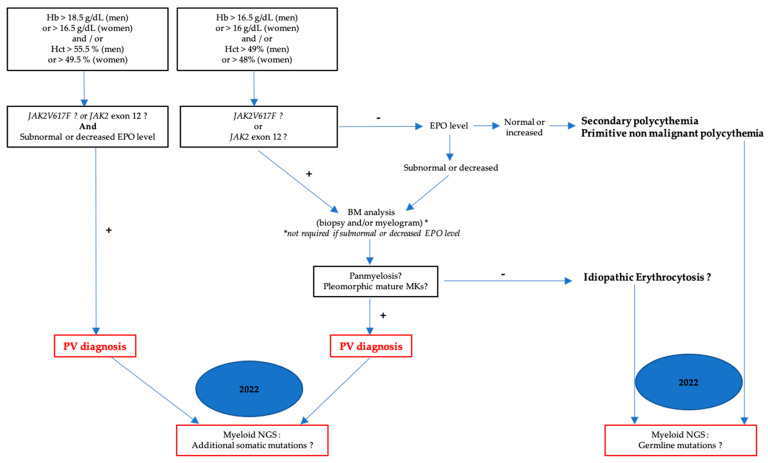
Diagnostic algorithm for PV.

**Figure 4 cells-12-00946-f004:**
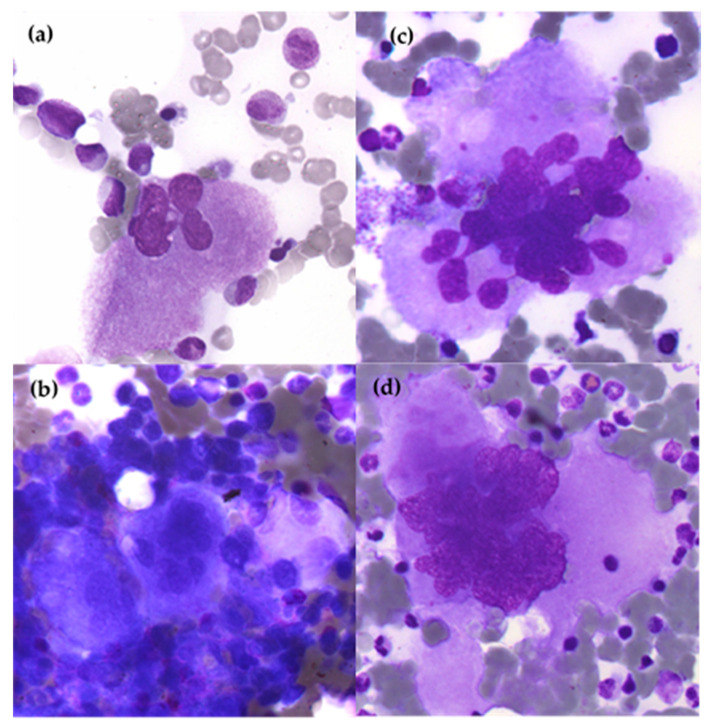
Cytological aspects of megakaryocytes in ET BM compared to physiological BM. (**a**) Physiological megakaryocyte in BM; (**b**) Cluster formation of megakaryocytes in BM ET; (**c**,**d**) Giant megakaryocytes and hypersegmented nuclei with «staghorn-like» aspect in ET BM. (optical microscopy, MGG and 100× magnification).

**Figure 5 cells-12-00946-f005:**
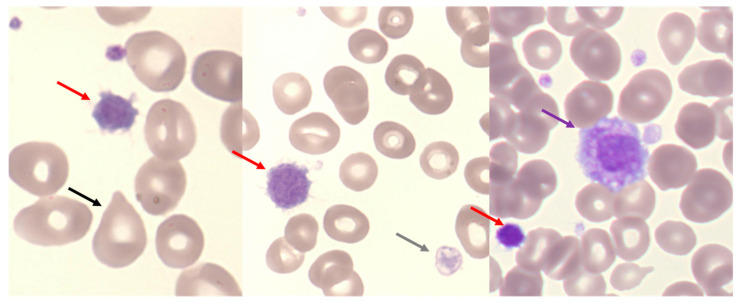
Blood smear in pre-PMF. Heterogeneity of platelets with macroplatelet (red arrows), giant platelet (purple arrow), degranulated macroplatelet (grey arrow) associated with teardrop-shaped RBC (black arrow). (Optical microscopy, May Grünwald Giemsa, ×100 magnification).

**Figure 6 cells-12-00946-f006:**
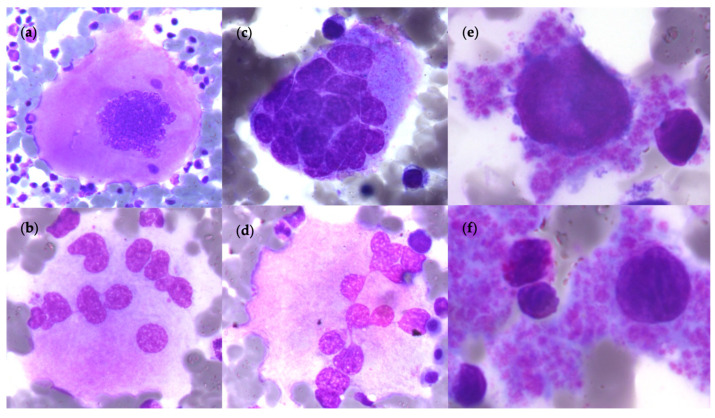
Morphological aspect of megakaryocytes in BM pre-PMF. (**a**) Atypical megakaryocytes with variable size (large to small) in pre-PMF: with hyperchromatic nuclei; (**b**–**d**) Enlarged megakaryocytes with variable lobulation; (**e**,**f**) Scattered small megakaryocytes with nude nuclei in platelet clusters. (Optical microscopy, May Grünwald Giemsa, ×100 magnification).

**Figure 7 cells-12-00946-f007:**
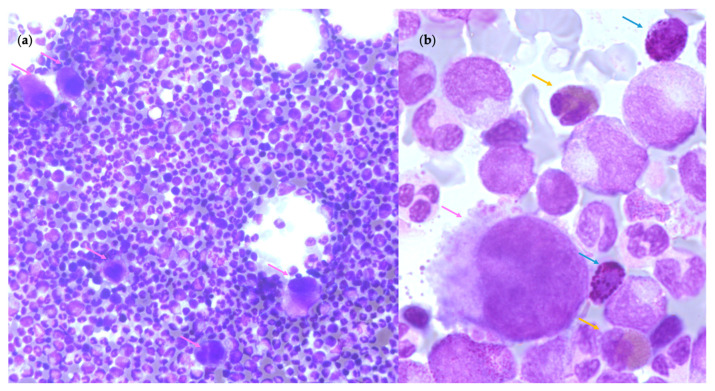
Bone marrow cellularity and morphological aspects of megakaryocytes in Chronic Myeloid Leukemia. (**a**) Granulocytic hyperplasia and small MKs (pink arrows) (optical microscopy, May Grünwald Giemsa (MGG), ×20 magnification) (**b**) A small MK and eosinophil (yellow arrows) and basophil (blue arrows) cells (optical microscopy, May Grünwald Giemsa (MGG), ×50 magnification).

**Figure 8 cells-12-00946-f008:**
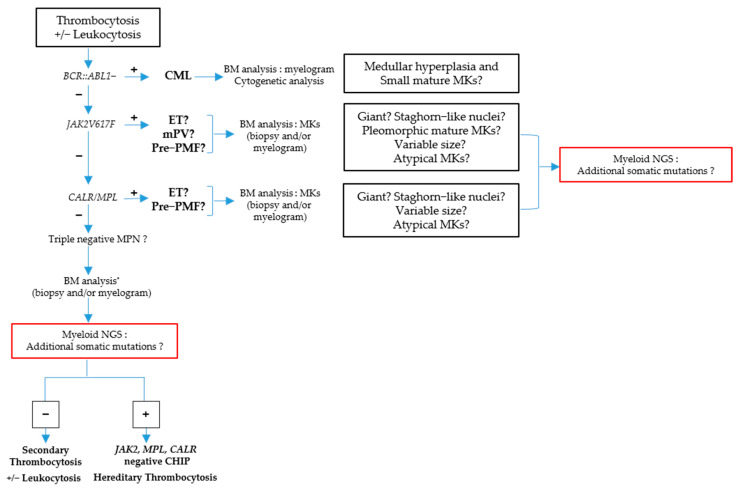
Algorithm diagnosis for CML, ET, prePMF.

**Table 1 cells-12-00946-t001:** Classification criteria for PV diagnosis.

WHO 2016 classification [18]	Major criteria: 1-Hb > 16.5 g/dL (men), >16.0 g/dL (women) or, Hct > 49% (men), >48% (women) or, increased red cell mass. 2-BM biopsy showing hypercellularity for age with trilineage growth (panmyelosis) including prominent erythroid, granulocytic and megakaryocytic proliferation with pleomorphic, mature megakaryocytes. 3-*JAK2V617F* or *JAK2* exon 12 mutation. Minor criterion: Subnormal serum EPO level

**Table 2 cells-12-00946-t002:** Somatic mutations and frequencies in MPN.

	PV	ET	PMF
	Driver mutations (Frequencies %)		
Signaling MPN driver	*JAK2V617F* (95%)	*JAK2V617F* (50–70%)	*JAK2V617F* (40–50%)
[33,48,54,55]	*JAK2* exon 12 (3%)		
	*MPL* (<1%)	*MPL* (3–15%)	*MPL* (3–15%)
	*CALR* (<1%)	*CALR* (25%)	*CALR* (25–30%)
	Additional mutations (Frequencies %)		
DNA methylation	*TET2* (9–16%)	*TET2* (4–5%)	*TET2* (7–17%)
[10,55]	*DNMT3A* (1–7%)	*DNMT3A* (5–10%)	*DNMT3A* (5–10%)
	*IDH 1/2* (2%)	*IDH1/IDH2* (3–6%)	*IDH1/IDH2* (3–6%)
Histone modification	*ASXL1* (7%)	*ASXL1* (4%)	*ASXL1* (18–35%)
[10,56]	*EZH2* (0–3%)	*EZH2* (0–3%)	*EZH2* (6–13%)
mRNA Splicing	*SF3B1* (3–5%)	*SF3B1* (3–5%)	*SF3B1* (5–7%)
[10,33]	*SRSF2* (0–3%)	*SRSF2* (0–3%)	*SRSF2* (8–17%)
	*U2AF1* (1–2%)	*U2AF1* (1–2%)	*U2AF1* (5–15%)
	*ZRSR2* (0–2%)	*ZRSR2* (0–2%)	*ZRSR2* (0–2%)
Transcription factors	*TP53* (1–3%)	TP53 (1–3%)	*TP53* (4%)
[10,54]	*IKZF1* (<3%)	*IKZF1* (<3%)	*IKZF1* (<3%)
	*RUNX1* (<3%)	*RUNX1* (<3%)	*RUNX1* (<3%)
Other signaling	*LNK/ SH2B3* (2–10%)	*LNK/ SH2B3* (3–6%)	*LNK/ SH2B3* (11%)
pathways	*CBL* (0–2%)	*CBL* (0–2%)	*CBL* (0–6%)
[10,33]	*NRAS*, *KRAS* (0–1%)	*NRAS*, *KRAS* (0–1%)	*NRAS, KRAS* (3–15%)

**Table 3 cells-12-00946-t003:** Classification criteria for ET diagnosis.

WHO 2016 classification [18]	Major criteria: 1-Platelet count ≥ 450 × 10^9^/L. 2-BM biopsy showing proliferation mainly of the megakaryocytic lineage with increased number of enlarged, mature megakaryocytes with hyperlobulated nuclei. No significant increase or left shift in neutrophil granulopoiesis or erythropoiesis and very rarely minor increase in reticulin fibers (≤grade 1). 3-No WHO criteria for other myeloid neoplasms. 4-*JAK2*, *CALR* or *MPL* mutation. Minor criterion: Presence of clonal marker or absence of evidence for reactive thrombocytosis

**Table 4 cells-12-00946-t004:** Classification criteria for pre-PMF and PMF diagnosis.

	Pre-PMF	Overt PMF
WHO 2016 classification [18]	Major criteria: 1-Megakaryocytic proliferation and atypia, without reticulinic fibrosis > grade 1, accompanied by increased age-adjusted BM cellularity, granulocytic proliferation and often decreased erythropoiesis. 2-No WHO criteria for other myeloid neoplasms. 3-*JAK2*, *CALR* or *MPL* mutation or, in the absence of these mutations, presence of another clonal marker or absence of minor reactive BM reticulin fibrosis.	Major criteria: 1-Megakaryocyte proliferation and atypia accompanied by either reticulin and/or collagen fibrosis. 2-No WHO criteria for other myeloid neoplasms. 3-*JAK2*, *CALR* or *MPL* mutation or other clonal marker or, absence of reactive bone marrow fibrosis.
	Minor criteria: 1-Anemia not attributed to a comorbid condition. 2-Palpable splenomegaly. 3-Leukocytosis ≥ 11 × 10^9^/L. 4-Elevated LDH (confirmed in two consecutive determinations). 5-Leukoerythroblastosis. *	

* only a criterion in overt PMF.

**Table 5 cells-12-00946-t005:** Cytomorphologic features for pre-PMF and PMF diagnosis.

	Pre-PMF	Overt PMF
CBC and blood smear	Thrombocytosis	Leukoerythroblastosis, thrombopenia/anemia
Myelogram/ BM biopsy	Megakaryocytic proliferation with atypia and large spectrum of shapes (abnormal maturation, hyperchromatic, hypo or hyperlobulated, irregular nuclei (bulbous, “cloudlike”), tight clusters Reticulin fibrosis ≤ grade 1 (pre-PMF) Reticulin and/or collagen fibrosis grade 2 or 3 (overt PMF) [15]	

**Table 6 cells-12-00946-t006:** Classification criteria for CML diagnosis.

	Accelerated Phase (AP)	Blast Phase (BP)
WHO 2016 classification [18]	Any 1 or more of the following: Hematological/Cytogenetic criteria or response to TKI criteria: - Persistent or increasing leukocytosis (>10 × 10^9^/L) unresponsive to therapy. - Persistent or increasing splenomegaly unresponsive to therapy. - Persistent thrombocytosis (>1000 × 10^9^/L) unresponsive to therapy. - Persistent thrombocytopenia (<100 × 10^9^/L) unrelated to therapy. - Basophils cells: ≥20% of the total in blood. - Blast cells: 10–19% of the total in blood or BM (<10% if lymphoblast). - Additional clonal chromosomal abnormalities in Philadelphia chromosome-positive cells at diagnosis that include “major route” abnormalities (second Ph, trisomy 8, isochromosome 17q, trisomy 19), complex karyotype, or abnormalities of 3q26.2. - Any new clonal chromosomal abnormality in Ph^+^ cells that occurs during therapy. Or Provisional response-to-tyrosine kinase inhibitor (TKI) criteria: - Hematological resistance to the first TKI (or failure to achieve a complete hematological response). - Any hematological, cytogenetic, or molecular indications of resistance to two sequential TKIs. - Occurrence of two or more mutations in *BCR::ABL1* during TKI therapy.	Myeloid blast cells ≥ 20% of total in blood or BM. Extramedullary proliferation of blasts.
WHO 2022 classification [19]	No criteria AP is omitted	Myeloid blast cells ≥ 20% of the total in the blood or BM. Extramedullary proliferation of blasts. Presence of increased lymphoblasts in PB or BM.

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
