# Peer review of "Cytological Diagnosis of Classic Myeloproliferative Neoplasms at the Age of Molecular Biology"

_cells, 2023, doi:10.3390/cells12060946_

Round 1

Reviewer 1 Report

This paper by Combaluzier el al summarizes current knowledge on how cytomorphological and molecular markers are currently integrated for the diagnosis of myeloproliferative neoplasms.

Overall, the review of the state of the art is accurate although some oversights in the description of some of the data are recognized. The take home message of the paper is clear and indicates that cytomorphological criteria are sufficient for an accurate diagnosis of the various MPNs. This take-home message is the reality of clinical practice in small centers.

What the cytological criteria fail to predict is risk stratification and identification of optimal therapy. Here is when molecular diagnosis came to help. This very important concept is insufficiently discussed by the paper. This is a serious weakness because by reading this paper clinicians active in the field who work in small hospitals with limited access to certified molecular diagnostic laboratories may not understand why they should do an effort to integrate cytomorphological with molecular diagnostic criteria. Another reason why this is a weakness is that the validation of molecular diagnosis for risk stratification and therapeutic choice is currently one of the most active areas of research in the field. As an example, it is already established that MPN patients with CARL mutations have a more favorable prognosis than those carrying the JAK2 or MPL mutations. Also, the order with which a patient acquires the driver and the secondary mutation appears to be relevant for prognosis with those patients who acquire the secondary mutations first having a greater risk to progress to leukemia than those who acquire the driver mutation first. All of this was not sufficiently discussed.

The authors may wish to decrease the amount of historical information they provide and focus on the diagnostic criteria implemented by WHO criteria lastly published. They may want then to expand the discussion on how the molecular diagnosis may assist with risk stratification, prognosis and therapeutic choices with the ultimate goals of personalized therapies.

How artificial intelligence and computer-based observations may change the histopathological assessment of the MPN in the future should at least be mentioned.

Minor comments

Figures. Arrows pointing to the major morphological features of the morphological data presented should be added to all the panels.

Table 2 appears redundant with respect to Table 1 and should be deleted.

Ref.16 appears incomplete.  

Only a limited number of citations has been published after 2021. This is a weakness in view of the wealth of information published on this subject in the last two years (as an example see Ryou et al, Leukemia 2022; https://doi.org/10.1038/s41375-022-01773-0).

Figure 3. is very difficult to read and appears in contradictions. Why if JAK2 mutations are present and EPO levels are down biopsy is not required while if EPO levels are down in the absence of JAK2 mutations, biopsy is required? In addition, in the presence of normal/high EPO levels, the presence of germline or acquired mutations should also be investigated.  

Table 5 appears redundant with Table 4 and could be deleted.

323. What it is meant 20-40%? Volume? Diameter? Please clarify.  

Figure 4 and 5. The quality of these figures must be improved.

Reviewer 2 Report

This review by Sophie Combaluzier et al summarized the morphological appearance of main BCR::ABL1-negative (PV, ET and PMF) and positive (CML) MPN. Besides, authors offered the association of phenotypic and molecular aspects. WHO has updated the diagnosis and classification of myeloproliferative neoplasms, which insists on the importance of association of quantitative data from the hemogram, cytological analysis as well as molecular data to increase the diagnostic specificity. This review well catered the new edition. What’s more, the content is essential in discussion about differential diagnosis and difficult or atypical MPN cases.

Here I have a few minor but important comments that need to be addressed.

1.     The manuscript only involves typical types of MPN (PV, ET, PMF and CML), It is suggested to change title to “Cytological diagnosis of classic myeloproliferative neoplasms at the age of molecular biology”.

2.     The part of “3. BCR::ABL1-POSITIVE MPN : Chronic Myeloid Leukemia (CML)” lack morphological figures.

3.     The molecular aspects of CML is insufficient, more molecular aspects and cytogenetic abnormalities should be involved.

Round 2

Reviewer 1 Report

My comments were properly addressed. No additional comments. Thanks